# Spinal Meningiomas: A Comprehensive Review and Update on Advancements in Molecular Characterization, Diagnostics, Surgical Approach and Technology, and Alternative Therapies

**DOI:** 10.3390/cancers16071426

**Published:** 2024-04-07

**Authors:** Danielle D. Dang, Luke A. Mugge, Omar K. Awan, Andrew D. Gong, Andrew A. Fanous

**Affiliations:** 1Department of Neurosurgery, Inova Fairfax Medical Campus, Falls Church, VA 22042, USA; danielle.dang@inova.org (D.D.D.); lukealan.mugge@inova.org (L.A.M.); omar.awan@inova.org (O.K.A.); andrew.gong@inova.org (A.D.G.); 2Department of Neurosurgery, Inova Alexandria Hospital, Alexandria, VA 22304, USA

**Keywords:** spine, meningioma, intradural extramedullary tumor, clear cell meningioma, radiotherapy, immunotherapy, laminectomy, laminoplasty, minimally invasive surgery

## Abstract

**Simple Summary:**

This study offers a comprehensive review of the current basic science and clinical literature concerning the diagnosis and treatment of spinal meningiomas in conjunction with illustrative case studies to emphasize up-to-date knowledge on molecular genetics, surgical resection, and alternative therapies.

**Abstract:**

Spinal meningiomas are the most common intradural, extramedullary tumor in adults, yet the least common entity when accounting for all meningiomas spanning the neuraxis. While traditionally considered a benign recapitulation of their intracranial counterpart, a paucity of knowledge exists regarding the differences between meningiomas arising from these two anatomic compartments in terms of histopathologic subtypes, molecular tumor biology, surgical principles, long-term functional outcomes, and recurrence rates. To date, advancements at the bench have largely been made for intracranial meningiomas, including the discovery of novel gene targets, DNA methylation profiles, integrated diagnoses, and alternative systemic therapies, with few exceptions reserved for spinal pathology. Likewise, evolving clinical research offers significant updates to our understanding of guiding surgical principles, intraoperative technology, and perioperative patient management for intracranial meningiomas. Nonetheless, spinal meningiomas are predominantly relegated to studies considering non-specific intradural extramedullary spinal tumors of all histopathologic types. The aim of this review is to comprehensively report updates in both basic science and clinical research regarding intraspinal meningiomas and to provide illustrative case examples thereof, thereby lending a better understanding of this heterogenous class of central nervous system tumors.

## 1. Introduction

Spinal meningiomas are typically intradural extramedullary tumors of the spine, which arise from arachnoid cap cells of the leptomeninges [1]. While relatively rare, accounting for approximately 1.2% of all meningiomas in the central nervous system (CNS), spinal meningiomas represent the most common primary spinal tumors in adults [1,2,3]. While these tumors have been traditionally believed to constitute a predominantly benign pathology, regardless of location in the neuraxis, evolving research depicts a more heterogeneous representation of underlying tumorigenesis mechanisms, molecular behavior, clinical outcomes, and recurrence patterns, as well as potential therapeutic avenues for tumor management. As medical and surgical technology concomitantly evolve along with significant improvements in radiographic and histopathologic analysis, such treatment options become more nuanced and complex.

New literature is emerging, which suggests that spinal meningiomas may differ from their intracranial counterparts in a number of pertinent ways, including tissue lineage and clonality, molecular characterization and pathways pertinent to meningiomagenesis, risk of recurrence, the utility of surgical grading systems, and adjuvant treatment strategies [2,4,5,6,7,8,9]. Nonetheless, regardless of location, ongoing challenges in meningioma research include the diagnostic integration of histopathology with molecular markers to accurately predict biological behavior, improvement in post-surgical neurological outcomes, and methods to decrease rates of recurrence. The current literature has a strong predilection for addressing such challenges in intracranial meningiomas with little consideration for the potential differences in those arising from the spine. Furthermore, spinal meningiomas tend to be combined with all intradural extramedullary lesions, which ultimately limits our collective understanding of this unique and heterogenous pathology.

This publication provides a comprehensive review of the current and evolving knowledge of spinal meningiomas spanning both basic science and clinical research. As such, it offers an updated understanding of the epidemiology, radiographic appearance, histopathologic diagnosis, molecular biology, surgical approaches, alternative medical therapies, and long-term outcomes related to this disease entity. Furthermore, it provides illustrative case studies to contextualize the stereotypical case and integrate this knowledge.

## 2. Methods

A review of the scientific literature pertaining to advancements in knowledge in the following domains was performed for spinal meningiomas through July 2023: epidemiology, clinical presentation, neuroimaging, histopathology, molecular genetics, surgical treatment and clinical outcomes, and adjuvant therapies. Keywords used in multiple databases included “spine”, “spinal”, “meningioma”, “intradural extramedullary tumor”, “clear cell meningioma”, and “primary spinal tumor”. Bibliographies of the relevant literature were iteratively reviewed for inclusion until each topic was comprehensively analyzed with a focus on studies and reviews performed within the last five years. Studies were not restricted by type but were restricted to the English language.

For the purpose of this review, we refer to “spinal” tumors as lesions occurring in any compartment within the intraspinal space, including the osseous structures, extradural soft tissue, leptomeninges, and/or neural elements in contrast with “spinal cord” tumor, which should only refer to the spinal cord parenchyma or intramedullary space. When discussing the classification of tumors that arise in the same anatomical space as the prototypical spinal meningioma, the term “intradural extramedullary” spinal tumor is further specified.

## 3. Epidemiology and Clinical Presentation

Tumors arising from the spinal meninges account for 1.7% of all tumors for all ages according to a recent epidemiological study for 2014–2019 [10]. Among spinal neoplasms, those occurring in the intradural extramedullary compartment account for approximately 20% of all lesions in adults and 35% in children, with meningiomas (37.5%) and ependymal tumors (17.6%) comprising the most common intradural extramedullary histopathologies in adults and children, respectively [10,11,12,13]. In the pediatric population, spinal meningiomas consist of 5% of tumors found within the intradural space [10]. Rare cases of isolated extradural spinal meningiomas have been reported [14], although these are typically coupled with the presence of concomitant intradural disease in patients with familial disease.

Spinal meningiomas are more common in elderly women, with a male-to-female ratio of 1:2.7 and with a peak incidence at presentation in the seventh decade of life [13,15,16]. When arising in the pediatric population, however, male overrepresentation is evident [15]. Racial distribution appears predominantly White, although most studies are based on the distribution of care-seeking patients in the United States and remain significantly limited [15]. The only known risk factors for meningiomatosis include prior exposure to ionizing radiation [17,18] and the diagnosis of neurofibromatosis type 2, a genetic condition characterized by multiple inherited meningiomas, schwannomas, and ependymomas [12,19]. While the presence of steroid hormone receptors promotes tumor growth [20], the characterization of oral contraceptive use and pregnancy as risk factors for meningiomagenesis is not supported in large-scale population studies [21,22,23]. Meningiomas below the craniocervical junction occur most frequently within the thoracic spine (80%) followed by the cervical spine (15%) in adult women, whereas the location tends to be more evenly distributed in adult men [11]. Pediatric series report a higher proportion of cervical tumors [15,24]. No consistent consensus exists with regard to the predominance of axial location, with some studies suggesting that lateral overrepresentation may be attributed to subgrouping with ventral or dorsal compartments [15].

Clinically, spinal meningiomas are often characterized by an insidious and progressive course beginning with non-specific focal back pain. Across 50 studies, the most common presenting symptoms included motor dysfunction (92%), sensory dysfunction (78%), and pain (76%), as well as sequelae of spinal cord compression, like gait disturbance (42%) and bowel and bladder dysfunction (28%) [15,25]. Such symptoms seem to predominate in elderly patients, thoracic location, and tumor occupancies of over 64% of the spinal canal [15,25]. As tumor growth continues, spinal cord compression increases, resulting in signs of myelopathy, such as spastic weakness and hyperreflexia [26]. The insidious presentation of these tumors is further highlighted by the fact that 21% to 53% of patients are unable to ambulate at the time of diagnosis [2,3,7,27,28,29,30,31]. Focal neurological deficits ultimately map to the myotomes and dermatomes arising from stretched nerve roots and ascending or descending spinal tracts compressed by the growing lesion. However, studies have failed to show a consistent correlation of specific symptoms with axial location [15,25,32,33]. Pain appears to be more common in cervical tumors, while sensorimotor, bowel, and bladder dysfunction are associated with thoracic lesions [25,34]. Regardless of craniocaudal or axial location, symptoms typically progress over a period of six months to three years, and diagnosis typically lags the onset of clinical symptoms by an average of 13.6 months and 8 months for adults and children, respectively [15,35]. Some studies suggest that female sex and a higher histological tumor grade correlate with shorter symptom durations, although these findings have not been validated [7,36].

In the literature, the severity of neurologic dysfunction in patients with spinal meningiomas is often classified by the modified McCormick grading scale, which spans from the complete absence of symptoms (grade 1) to being non-ambulatory with bilateral upper extremity weakness (grade 5) [37]. The Frankel classification is another system adapted to establish the severity of neurological deficit as a result of epidural spinal cord compression, ranging from Frankel grade A (no sensorimotor function below the affected level) to Frankel grade E (no deficits) [38]. One study found that the Frankel scale weakly correlated with the extent of spinal cord compression and served as a predictor for postoperative neurological deterioration [39]; however, no large-scale studies have been performed to compare and contrast the utility of various clinical grading schemes such as those above.

## 4. Neuroimaging

The radiographic features of spinal meningiomas are best depicted through magnetic resonance imaging (MRI). These tumors are typically intradural extramedullary, well-circumscribed solid lesions [40]. They are generally iso- or hypointense on T1-weighted imaging and iso- or hyperintense on T2-weighted images, with homogeneous T1-weighted post-contrast enhancement [15]. One study found that lesions with higher T2-weighted signal intensity on MRI correlated with a soft consistency and were easier to debulk with an ultrasonic surgical aspirator [41]. A dural tail, or thickening of the peritumoral dura, can often be visualized and used to differentiate these lesions from other pathologies, such as schwannomas [36,42,43,44,45,46]. T2-weighted signal change within the spinal cord may be present in up to 68% of patients, signifying significant compression by the tumor [47]. Rarely, meningiomas can present as an “en plaque” lesion, with a broad-based morphology stretching along the dura and extending multiple vertebral levels [14]. These lesions often demonstrate peripheral calcifications along the dura [48].

Even rarer, accounting for less than two percent of all meningiomas, are primary extradural meningiomas, which can occur in the calvaria, scalp, orbit, paranasal sinuses, oropharynx, nasopharynx, soft tissue of the neck, skin, and boney vertebrae [49,50]. A subset of these tumors, termed “primary intraosseous osteolytic meningiomas (PIOM)”, constitute two-thirds of these isolated extradural meningiomas and are non-dural-based osteolytic lesions with or without adjacent hyperostosis, most commonly occurring in the calvarium; extremely rare instances occurring either as isolated intraosseous spinal meningiomas or as posterior fossa masses with extension into the boney cervical vertebrae have been reported [49,50,51,52]. In contrast to their cranial counterparts, typical intradural extramedullary spinal meningiomas rarely involve bony structures or penetrate the pia mater [53].

Additional diagnostic radiographic modalities may include computed tomography (CT) to assess for the presence of calcification and even positron emission tomography (PET) to determine the degree of tumor metabolism that can help differentiate between spinal meningiomas and schwannomas [44]. Calcifications on CT, which tend to match areas of MR hypointensity, also help favor a diagnosis of meningioma, although these are invariably present and not necessarily pathognomonic [36,44]. According to one study utilizing CT imaging, rates of calcification were lowest in the pediatric population [54]. Understanding the degree of calcification within a meningioma is important since calcifications increase the difficulty of tumor manipulation and removal of dural adhesion during surgery [40].

Once a spinal meningioma is suspected based on clinical and radiographic presentation, spinal angiography can be used to evaluate the lesion’s vascular supply and risk of spinal cord ischemia, particularly in lower thoracic lesions that occur near the artery of Adamkiewicz; the largest anterior segmental medullary artery that typically lies between T8 and L3 [55,56]. Additionally, spinal angiography may be used for preoperative embolization of hypervascular meningiomas to decrease tumor volume and the risk of intraoperative hemorrhage [56]. While meta-analyses have attempted to evaluate the optimal patient population likely to benefit from preoperative embolization of intracranial meningiomas, no such study has been performed for spinal tumors [57].

## 5. Origin, Histopathology, and Grading

The cells of origin for all meningiomas are believed to be the arachnoid cap cells of the neural crest or mesodermal origin [42,58]. Grossly, these tumors are usually round, well circumscribed, and attached to the dura. Spinal meningiomas typically arise laterally in the leptomeningeal sheaths at the outlet of spinal nerve roots and separate readily from the spinal cord. Less commonly, they can also arise ventrally or dorsally from mesodermal fibroblasts [53]. In the case of intraosseous meningiomas, several mechanisms have been posited for their rare, isolated extradural origin in the bone, including defects of neural crest migration, origin of arachnoid cap cells from nerve sheaths protruding from skull and/or spinal foramina, ectopic arachnoid granulations, trauma, intracranial hypertension causing movement of arachnoid cells, and origin from undifferentiated mesenchymal cells [49]. The World Health Organization (WHO) Classification of Tumors of the Central Nervous System (CNS) subdivides meningiomas into grades 1, 2, and 3 based on histopathologic features of malignancy, such as mitotic activity [22,58]. This grading has been shown to correlate with the risk of recurrence [8,59].

The overwhelming majority of spinal meningiomas (95.5%) are benign WHO grade 1 tumors with minimal mitotic activity and a recurrence rate of less than 25% [15,58]. WHO grades 2 and 3 constitute 4.3% and 0.4% of the remaining tumors, respectively [15,58]. Of the 15 different histopathological subtypes of meningiomas depicted in the fifth edition of the WHO classification [58,59], the most common subtypes of spinal meningioma are all grade 1 lesions, including psammomatous (40.1%), meningothelial (34.0%), transitional (13.9%), and fibrous (8.6%) meningiomas [15,58,60]. The psammomatous subtype typically demonstrates numerous psammoma bodies with few meningothelial cells, which usually results in large, calcified masses. This subtype is most common in the thoracic spine of older women. Meningothelial meningioma is typified by meningothelial whorls, syncytial cells with round uniform nuclei, intranuclear pseudoinclusions, and immunohistochemical staining for markers like somatostatin receptor 2a (SSTR2A), epithelial membrane antigen (EMA), and vimentin and progesterone receptors [58]. Fibrous subtypes of meningioma contain spindle cells with thick bundles of collagen, often resembling schwannomas or solitary fibrous tumors [53,58]. Transitional meningiomas are a mixed subtype that contain both meningothelial and fibroblastic features [58,59]. Angiomatous, metaplastic, lymphoplasmacyte-rich, microcystic, and secretory meningiomas are rare tumors that account for the remaining 2% of grade 1 spinal meningiomas [15,58].

Grade 2, or atypical, meningiomas demonstrate increased mitotic activity (4–19 mitoses per 10 high power microscope fields of 0.16 square millimeters), pial invasion, or have three of the following five atypical morphological features, including increased cellularity, small cells with a high nucleus to cytoplasm ratio, prominent nucleoli, sheeting, or spontaneous foci of necrosis [58,59,61]. Two histopathologic subtypes comprise these higher-grade lesions based on microscopic morphological appearance, including clear cell and chordoid, which comprise 29.4% and 3.9% of grade 2 meningiomas, respectively [15,58]. The remaining 66.7% of grade 2 meningiomas are non-morphologically defined by the above criteria [15,58]. These lesions can have recurrence rates of up to 50% [62]. Grade 2 spinal meningiomas commonly present earlier in life, with a greater incidence found in pediatric populations compared to adults [63,64]. Of note, the clear cell subtype, which demonstrates a sheeting architecture with round clear cells, is especially common in the cauda equina and has a particularly high recurrence rate given its ability to seed cerebrospinal fluid (CSF) [15,58].

Grade 3, or anaplastic, meningiomas are very rare lesions characterized microscopically by frank anaplasia and more than 20 mitoses per 10 high-power microscope fields of 0.16 square millimeters [58,61]. Given diagnostic advancements using molecular genetics (see the Molecular Genetics section), the updated definition of grade 3 (anaplastic) meningiomas now extends beyond traditional morphological subtypes and non-morphological criteria to include the presence of a telomerase reverse transcriptase (TERT) promotor and/or homozygous CDKN2A and/or CDKN2B deletions. Common histopathological subtypes that often result in either a grade 2 or grade 3 rating based on these criteria include papillary and rhabdoid meningiomas. However, these subtypes are no longer automatically grade 3 meningiomas based solely on their morphological appearance as encountered in past classification schemes [58,59]. Non-morphologically defined grade 3 (anaplastic) meningiomas comprise about 71% of all grade 3 spinal meningiomas yet, as an entity, are still less frequently observed in the spine compared to the cranium [15,58,65]. When anaplastic spinal meningiomas occur, they have been reported primarily either as primary lesions with concurrent NF2 diagnosis [66] or as metastases from intracranial meningiomas undergoing “malignant transformation” via seeding of the cerebrospinal fluid [67,68]. These lesions portend a poor prognosis with up to 100% recurrence rates and a five-year survival rate of less than 50% [62].

Grading based on histopathology alone is ultimately limited due to inter-observer variability in histological assessment, potential under-sampling of tumors with histologic and molecular heterogeneity, and an incomplete understanding of tumor biology represented solely by tissue appearance and architecture [69]. As a result, discrepancies persist between previous WHO grading schemes, tumor recurrence, and prognosis. Thus, a number of recent publications have proposed alternative grading systems based on molecular genetics, which can model and predict recurrence and progression-free survival more accurately than the current WHO grading system [22,69].

## 6. Molecular Genetics

With recent advances in molecular genetics and improved knowledge of epigenetic mechanisms, molecular profiling has become critical to our understanding of meningiomagenesis. Yet, most of the knowledge regarding gene targets, molecular characterization, and proteomic pathways of meningiomagenesis stems from studies centered on intracranial meningiomas [58]. Meningiomas can generally be grouped genetically into two subtypes. Over 50% of meningiomas have sporadic loss of chromosome 22q12.2 between the loci of D22S212 and D22S32, which encode the neurofibromatosis 2 gene [70,71]. This genetic subset is more likely to progress to a higher WHO grade given its increased propensity to develop TERT promotor mutations and CDK2NA/B deletions, as is now reflected in the updated WHO definition of grade 3 meningiomas [70]. Given the extreme rarity of anaplastic spinal meningiomas and the recent diagnostic integration of histopathology with molecular genetic analysis in meningioma classification, no reports currently exist to propose the incidence of these particular molecular profiles. To date, only one case discusses the diagnosis of a pediatric spinal meningioma in the context of a new neurofibromatosis type 2 (NF2) diagnosis with multiple CNS lesions whereby the dominant intracranial tumor was surgically resected and histopathologically confirmed as a meningothelial grade 3 meningioma with chromosome 22 loss and without TERT mutation and CDKN2A/B deletions; however, this genetic profile was never separately confirmed for the intraspinal tumor [66].

The second genetic subtype of lesions that lack chromosome 22 mutations but include other mutations that cause sporadic meningiomas include TNF receptor-associated factor 7 (TRAF7), Kruppel-like factor 4 (KLF4), AKT serine/threonine kinase 1 (AKT1), DNA-directed RNA polymerase II polypeptide A (POLR2A), phosphatidylinositol 3-kinase catalytic subunit A (PIK3CA), smoothened frizzled class receptor (SMO), TERT, SWI/SNF-related matrix-associated actin-dependent regulator of chromatin subfamily E member 1 (SMARCE1), matrix metallopeptidase-9 (MMP-9), and chromosome 1p and/or 9p alterations [58,62,72,73,74,75,76]. Predisposing genetic conditions associated with the presence of spinal meningiomas include NF2, schwannomatosis, multiple endocrine neoplasia type 1 (MEN 1), and, rarely, neurofibromatosis type 1 (NF1) and Von Hippel Lindau (VHL) syndrome [62,77,78,79].

Previously, attempts have been made to correlate these mutations with histopathologic subtypes, locations of occurrence, and recurrence. More recently, stratification by molecular taxonomy has been considered an improved method to characterize tumor biology and predict clinical outcomes. Thus, recent efforts have introduced the integration of whole-exome sequencing, DNA methylome analysis, and mRNA sequencing into one unified analysis to create various tumor profiles [62,70,80,81]. While these integrated classes may eventually prove relevant for intraspinal meningiomas, to date, such analyses have been restricted to intracranial lesions, and the differences in tumor biology between spinal and cranial meningiomas remain unclear.

Meningiomas with a predilection for the spine are typically found in patients with multiple meningiomas, such as those who harbor a germline mutation in NF2. In these patients, the benign psammomatous histologic variant predominates, and spinal meningiomas co-exist with identical intracranial lesions [82]. First identified in a group of individuals with familial multiple spinal meningiomas without NF2 mutations, clear cell meningioma represents a rare malignant histologic subtype with a greater tendency to metastasize and with pathognomonic mutations in SMARCE1 [58]. SMARCE1 is a large ATP-dependent chromatin remodeling complex, which is responsible for stabilizing the nucleosome and allowing for the activation of normally repressed genes [83,84]. SMARCE1 mutations are typically mutually exclusive from other recurrent mutational events implicated in meningiomagenesis [83].

Outside of these well-defined clinical entities, the most consistently reported genetic hallmark of spinal meningiomas is the complete or partial loss of chromosome 22 with a tendency to originate from a single cell clone, in contrast to intracranial tumors, which often harbor multiple cells of origin [4,85]. Additional cytogenetic discoveries include the loss of 1p, 9q, and 10q and the gains of 5p and 17q chromosomes, which tend to be more frequently observed in the atypical and anaplastic subtypes [85]. Out of a differential expression of 1555 genes in intracranial and intraspinal meningiomas, 35 genes involved in transcription and intra- and extracellular signaling were found to be more highly expressed in the spine [4]. Spinal meningiomas also frequently demonstrate an upregulation of the progesterone receptor and increased MMP-9 expression, amounting to 86% and 46% of cases in one study, respectively [86]. Unlike intracranial meningiomas, however, increases in MMP-9 expression do not yield higher tumor proliferation in spinal meningiomas and may instead help predict relapse in the absence of progesterone expression [86]. While the advent of molecular and genomic profiling is beginning to impact the management of intracranial meningiomas, the application of such technology to primary spinal tumors has been lagging, and large-scale sequencing studies of spinal meningiomas are now necessary [87]. Finally, while molecular characterization has proven pertinent to the biological determination of meningioma progression and recurrence, some studies still suggest that factors related to surgery rather than genomic profiling confer the greatest contribution to recurrence, a supposition that again derives from intracranial meningioma research [88].

## 7. Surgical Resection

Safe, maximal surgical resection is the preferred treatment of choice for spinal meningiomas, regardless of patient demographics and tumor histopathology [89]. Successful operative intervention begins with preoperative consideration of patient selection, radiographic tumor characterization, perioperative medication use, and availability of ancillary surgical technology. While no specific studies have evaluated the perioperative administration of corticosteroids in spinal meningiomas, their utility is well-established for aiding in the resection of intracranial tumors through the minimization of peritumoral edema and modulation of the blood–brain barrier, leading to the preservation of neurological function postoperatively [90].

A myriad of technological advances have been introduced to enhance the resection of spinal meningiomas and minimize iatrogenic injury to neural tissue. The binocular surgical microscope was the first and most obvious addition, which enhanced visualization and enabled the use of microsurgical techniques [91]. More recently, the extracorporeal telescope, also known as the exoscope, has become a valuable alternative to the traditional microscope and offers benefits like enhanced magnification and three-dimensional visualization [92,93]. The intraoperative ultrasound represents another useful imaging modality, which permits real-time visualization of the intradural meningioma prior to the durotomy, thus guiding the extent of dural opening [94]. Ultrasonic tumor aspirators have increased the debulking efficiency, particularly in calcified lesions, and have been shown to minimize bleeding through their ultrasonic cavitation technology, while conferring minimal risk to the surrounding tissues [11,95,96,97]. Furthermore, intraoperative neuromonitoring with somatosensory (SSEP) and motor-evoked potentials (MEPs) has favorable sensitivity and specificity and may prove helpful in mitigating the risk of iatrogenic neurological injury [98]. Nonetheless, the available evidence for its utility in spinal meningiomas is lacking and confounded by the inclusion of nerve sheath tumors [99], with present conclusions demonstrating no clear improvement in postoperative outcomes [89]. Regardless of the surgical approach chosen, the employment of these various technologies contributes to successful and safe gross total resection of spinal meningiomas in up to 98% of cases [2,3,7,27,28,29,30,35,100].

### 7.1. Surgical Principles

Surgical approaches for the resection of dural-based spinal tumors span a variety of options and techniques but are all unified by one primary goal: to obtain adequate exposure that permits maximal visualization of the tumor with concomitant minimization of transgressing or retracting normal neural tissue [101]. The posterior midline approach is the most frequently used method described in the current literature followed by the posterolateral approach; the latter is typically reserved for high cervical tumors [89]. Anterior approaches, including a transoral approach to a high subaxial cervical meningioma, were reported in sporadic case series and are otherwise relatively rarely indicated [89].

To obtain access to the spinal canal in a posterior or posterolateral approach, laminectomy or modifications thereof are most commonly performed. Alternatives include hemilaminectomies, laminotomies, and laminoplasties. Additional resection of the posterior column elements in the form of facetectomies, costotransversectomies, and pedicle osteotomies have also been described. The decision between mono- and multi-segmental laminectomy versus osteoplastic laminotomy and the reconstruction of the posterior column has a number of implications [2,102]. Firstly, laminoplasty is associated with a mean increase in kyphosis of three degrees in the cervical spine [103]. This raises valid concerns about the integrity of posterior element reconstruction and its impact on the development of spinal deformity. On the other hand, laminoplasty is also associated with decreased CSF leak and hospital length of stay, indicating that reconstruction has implications beyond simple structural integrity [104]. Hemilaminectomy without dural resection has been also reported in cases of small spinal meningiomas [105,106].

Minimally invasive surgery (MIS) approaches have garnered burgeoning interest in intradural spinal tumor surgery to minimize tissue trauma, and while poorly defined as a group, are all related through preservation of structural integrity to decrease morbidity [89]. Examples of techniques considered to constitute MIS approaches in the literature include open hemilaminectomy, decreased length of the skin incision, the Saito method for dural splitting, dural coagulation to avoid radical resection, and use of expandable tubular retractors for myofascial preservation [2,33,34,105,107,108,109,110,111]. MIS approaches that seek to preserve the posterior column or avoid myofascial disruption are ultimately limited when there is foraminal involvement or when the tumor spans multiple vertebral levels [112]. Studies evaluating open versus MIS approaches for the treatment of intradural extramedullary tumors, including meningiomas, demonstrate no difference related to the extent of resection, but MIS techniques are associated with decreased intraoperative blood loss and are better tolerated in elderly patients [113,114,115]. The benefits of MIS for spinal meningioma resection with regard to operative duration and hospital length of stay remain poorly studied [89].

While the tenet of meningioma surgery is safe, maximal tumor resection to include the involved dura, surgery for spinal meningiomas differs from that of their intracranial counterparts. Radical resection of the dura in the spine significantly increases the risk of postoperative CSF leak and is, therefore, often avoided [2,110]. In fact, most studies favor dural coagulation at the attachment of the tumor to the dura rather than radical removal [111,116], resulting in a Simpson grade 2 resection [117]. This modification is particularly true when the tumor involves the ventral aspect of the spinal canal, where limited exposure and visualization restrict the ability to perform a direct repair or duraplasty. In fact, the reported rates of radical dural resection for spinal meningiomas are only between 14% and 58% [3,7,30,35]. Nonetheless, employing this technique has the theoretical risk of higher progression and recurrence rates secondary to the presence of residual tumor cells in the non-resected dura. An alternative method is the Saito technique, whereby the dura is effectively separated into two layers for resection of the inner layer with the tumor, while the outer layer is left intact to achieve a direct dural repair [111]. Nonetheless, lower rates of recurrence have not been demonstrated with this approach when compared with dural coagulation [111]. Given these considerations, the subsequent utility of the Simpson grading scale, arguably the most utilized system to predict recurrence based on the extent of resection in intracranial meningiomas, likely requires reconsideration in the context of dural-based spinal tumors [9,118,119]. To date, alternative grading scales to quantify the extent of resection in spinal meningiomas remain inconsistent, subjective, and non-standardized [89]. While intraoperative techniques and technologies may alter morbidity by improving the extent of safe resection, the aforementioned advances in molecular genetics likely remain the next frontier to better understand the biological behavior of spinal meningiomas and their risk for recurrence after surgery.

### 7.2. Challenging Cases

Three types of spinal meningiomas present technical challenges related to surgical resection: en plaque tumors, calcified lesions, and meningiomas in a ventral location. En plaque spinal meningiomas, while rare, are predominantly extradural and significantly adherent to the surrounding tissues, often circumferentially encasing the thecal sac. Gross total resection of en plaque meningiomas is not always feasible, although good clinical outcomes may still be achieved when adequate spinal cord decompression is performed [64]. In such cases of subtotal resection, close observation and follow-up are necessary.

Highly calcified meningiomas also present with increased tissue adherence as well as a gritty bulkiness that precludes facile manipulation to avoid iatrogenic neural injury. As a result, the degree of meningioma calcification has been shown to be inversely related to post-surgical neurological status outcomes [120]. This is particularly true when eloquent tissue is involved, such as in the thoracic or cervical spines, where significant thecal sac or spinal cord manipulation can be detrimental. In some studies, calcified tumors have been associated with longer operative durations and greater volumes of blood loss [42]. Of all spinal meningiomas, calcified meningiomas have a lower rate of gross total resection, averaging approximately 5%, with an increased risk of poor neurological outcomes [2,30].

While relatively uncommon and representing only 9% of all spinal meningiomas, ventral meningiomas are another group that warrants special surgical approach planning [2]. Due to the difficulty of accessing such tumors around the spinal cord from a posterior or posterolateral position, gross total resection to include dural excision is usually not feasible [32]. As a result, recurrence rates for ventral spinal meningiomas are higher than those for spinal meningiomas located more dorsally. In fact, ventrally located tumors account for 62% of all recurrent spinal meningioma cases [32].

### 7.3. Complications and Clinical Outcomes

The overall complication risk following surgery for spinal meningiomas is approximately 7.4%, with CSF leak being the most frequent complication followed by wound infections, wound revisions, new neurological deficits, and hematomas [89]. The most common cause of perioperative death in such surgeries is venous thromboembolism. Increased risk of complications has been associated with age greater than 70 years, ventral tumor location, tumor calcification, surgery for recurrence, obesity, longer operative durations, and surgeon inexperience [2,31,34,42,109,121].

Preoperative neurological status and longer time to surgery are the only factors found to predict postoperative functional outcomes following surgery for spinal meningiomas [89]. Several factors have also been associated with good functional outcomes following surgical resection of spinal meningiomas, including but not limited to posterior/lateral location, a location below C4, a patient age less than 60 years, and a relatively short duration of preoperative symptoms [7]. Gross total resection certainly results in good functional outcomes, and as many as 80% of patients are ambulatory at one year postoperatively [27].

In addition to optimizing immediate postoperative outcomes, efforts to minimize recurrence represent an active area of investigation for meningiomas in all locations of the neuraxis. The pooled recurrence rate for surgically treated spinal meningiomas is 6.0% at an average follow-up time of 62.9 months across all studies [89]. The pooled average time to recurrence is 59.8 months, indicating that the length of reported follow-up is insufficient to detect true recurrence rates [89]. While suggested risk factors for recurrence are diverse and span a variety of studied metrics with contradictory results, only male sex, WHO grades 2 and 3, and extent of surgical resection following Simpson grades 3, 4, or 5 correlate with significantly higher recurrence rates [89]. When these tumors recur, reoperation remains a safe and effective option [63].

## 8. Illustrative Clinical Case Studies

The following case studies represent the stereotypical type of patient, clinical presentation, histopathology, and treatment regime for spinal meningiomas in the cervicothoracic spine as discussed above.

### 8.1. Case One: Dorsolateral Thoracic Meningioma

A 68-year-old woman with a past medical history of fibroids, Hashimoto thyroiditis, osteopenia, and obstructive sleep apnea (OSA) presented with chronic mid-back pain, progressive sensory disturbance, and ambulation difficulty for six months. Neurological examination revealed a T9 sensory level and lower extremity hyperreflexia. Spinal MRI revealed a 2.1 × 1.4 × 1.2 cm intradural extramedullary lesion arising within the dorsal region of the canal at T7–T8 (Figure 1A–D). The spinal cord was displaced ventrally. The patient underwent laminectomies at T7–T8 for tumor resection with intraoperative ultrasound guidance. Intraoperative utilization of the microscope and neuromonitoring facilitated the identification of nerve roots, which were wrapped around the tumor requiring gentle displacement to enable safe tumor resection (Figure 1E,F). Bipolar cautery of the adherent dural attachment completed gross total resection of macroscopic disease, which was confirmed on postoperative imaging. Histopathology confirmed the diagnosis of a WHO grade 1 psammomatous meningioma. The patient demonstrated no postoperative complications and experienced a return to normal ambulation and resolution of myelopathy at follow-up. No recurrence of the meningioma has been demonstrated at 18 months follow-up.

### 8.2. Case Two: Ventrolateral Cervical Meningioma

A 73-year-old woman with a past medical history including a pituitary macroadenoma status post resection two years prior to presentation on hydrocortisone replacement, fibromyalgia, non-insulin-dependent diabetes mellitus, hepatitis, hyperlipidemia, hypothyroidism, and OSA presented with chronic cervicalgia, right-side radicular pain, and loss of dexterity for two years. Neurological examination was notable for interosseous and grip weakness bilaterally, a right-sided Hoffman’s reflex, and diffuse hyperreflexia. Spinal MRI revealed a 2.0 × 1.5 × 1.0 cm intradural extramedullary lesion arising within the ventrolateral region of the canal at C4–C6 (Figure 2A–D). The tumor extended into the right neural foramen, and the spinal cord was displaced dorsally. The patient underwent C4–C5 laminectomies for tumor resection with intraoperative ultrasound guidance. Intraoperative utilization of the microscope and neuromonitoring facilitated a safe working corridor to maneuver the tumor from its ventral position and in between nerve roots for resection (Figure 2E,F). Bipolar cautery of the adherent dural attachment completed gross total resection of macroscopic disease, which was confirmed on postoperative imaging. Histopathology confirmed the diagnosis of a WHO grade 1 meningioma without a histological subtype reported. The patient demonstrated no immediate postoperative deficit with eventual resolution of weakness and myelopathy. No recurrence of the meningioma was demonstrated at 12 months follow-up.

### 8.3. Case Three: Lateral Thoracic Meningioma

A 78-year-old woman with a past medical history of osteoarthritis, hyperlipidemia, and essential hypertension presented with mid-back pain and bilateral leg pain for three years, previously mitigated with epidural steroid injections. Neurological examination revealed diffuse hyperreflexia, inability to tandem walk, a right-sided Hoffman’s reflex, and palmar dysesthesias. Thoracic spine CT showed a hyperdense, concentric mass within the spinal canal at T8–T9 with intralesional calcifications in the posterolateral margin with adjacent hyperostosis of the right lamina, suspicious for meningioma (Figure 3A). Spinal MRI confirmed a 1.2 × 1.0 × 0.9 cm intradural extramedullary lesion arising at the right lateral aspect of the canal at T8–T9 in the setting of Scheuermann’s kyphosis (Figure 3B–E). The spinal cord was displaced to the contralateral side. The patient underwent T7–T9 laminectomies for tumor resection with intraoperative ultrasound guidance. Given the tumor’s lateralized position within the canal, medial facetectomies were performed on the right side for optimal visualization and tumor resection. A large thoracic nerve root was attached to the dorsal surface of the tumor as it traversed toward the foramen and was gently dissected away under microscopic guidance (Figure 3F). Bipolar cauterization of the adherent dural attachment completed gross total resection of macroscopic disease (Figure 3G), which was confirmed on postoperative imaging. Histopathology confirmed the diagnosis of a WHO grade 1 psammomatous meningioma with immunohistochemistry staining positive for epithelial membrane antigen (EMA) and progesterone and negative for S100. The patient demonstrated no postoperative deficit and experienced improved ambulation and balance. No recurrence of the meningioma was demonstrated at 12 months follow-up.

## 9. Adjuvant Therapies

Radical surgical resection is the mainstay of treatment of symptomatic spinal meningiomas and has been shown to improve quality of life postoperatively [122,123,124]. Despite the clear benefits of surgery, suggested indications for the use of nonsurgical treatment in the literature center on higher WHO grades, tumor recurrence, subtotal surgical resection, poor surgical candidacy due to increased risk of perioperative complications, NF2 disease, and the presence of multiple tumors [15]. In fact, nonsurgical treatments used as a primary, postoperative adjuvant, or salvage therapy were reported in 34.2%, 34.9%, and 10.4% of patients in a recent systematic review of the literature and included fractionated radiotherapy, stereotactic radiosurgery, chemotherapy, receptor Tyrosine Kinase inhibitors, brachytherapy, or a combination thereof [15]. However, the literature examining the use of adjuvant systemic therapy in spinal meningiomas remains highly variable in terms of treatment timing, indication, dose regimen, and outcomes. Therefore, conclusions regarding their utility in the optimal population continue to be limited and insufficient.

### 9.1. Radiotherapy

Radiation therapy has the potential to improve outcomes in patients unable to undergo surgical resection [125]. While the current treatment algorithms are well established for typical and low-grade intracranial meningiomas, few adjuvant therapies are available to guide treatment in atypical cases, and their associated protocols are currently poorly defined, particularly for spinal meningiomas [126]. Additionally, large-scale studies reporting recurrence rates are nonexistent, and those presenting data on survival outcomes are subjected to bias from confounders like age and genetic syndromes [60,127]. Radiotherapy is usually used as an adjuvant treatment modality following surgery, fractionated radiation therapy has been shown to decrease the rates of recurrence and to improve postoperative pain [128,129]. Across eight studies utilizing conventional radiotherapy after surgery, treatment failure defined by tumor recurrence was observed in 34.6% of tumors [15]. However, in cases where patients are poor surgical candidates, whether due to extensive medical comorbidities, significant tumor extension beyond the intraspinal compartment, multiple tumors, or for those who decline surgery, the primary use of radiotherapy alone is a viable option to delay or halt the progression of symptoms [127]. Reported complications related to radiation treatment include arachnoiditis, radiation necrosis, radiation-induced myelopathy, nausea, panic attacks, and constipation [15,130].

Studies examining the use of stereotactic radiosurgery (SRS) as the primary or adjuvant treatment of spinal meningiomas appear promising in terms of efficacy but are ultimately limited by small patient cohorts and short follow-up periods [131,132,133,134,135]. There are currently no consensus guidelines that provide recommendations for the radiation dosages when treating spinal meningiomas [136]. Treatment dosages typically range from 10 to 25 Gy [134,137,138,139,140]. Higher doses, namely, 15.9 Gy in a single fraction or 27.5 Gy distributed over five fractions, were shown to be both safe and efficacious, promoting tumor stabilization in 83% of patients with spinal meningiomas [135]. When SRS was used as a primary or adjuvant treatment, only 5% of 101 meningiomas reported in the literature experienced recurrence [15]; the one tumor in which multiple recurrences occurred was likely as a result of its origin from childhood radiation exposure [133]. In addition to its efficacy, SRS for the treatment of spinal meningiomas was also shown to be safe and without any delayed toxicity [141,142,143]. Between 90% and 100% of all spinal meningiomas treated with SRS remain symptomatically stable, with only 1% of such tumors demonstrating an asymptomatic increase in size [131,137,138]. Thus, while not the standard of care, SRS has demonstrated efficacy and can be considered in poor surgical candidates in patients with multiple spinal meningiomas or those with residual tumors following resection [144]. Since recent studies have demonstrated both its safety and efficacy, the rates of SRS utilization in the treatment of spinal meningiomas have been steadily increasing [127]. Although radiotherapy, including both fractionated external-beam radiation and SRS, can be used as a primary treatment with some benefit, its utility remains most efficacious as an adjuvant therapy following surgical resection. Finally, proton therapy has gained traction in its use to treat intracranial meningiomas due to a proton’s ability to achieve an improved dose conformation compared to photons, ultimately sparing healthy tissue [145]. However, there are currently no studies to date which have investigated its efficacy in spinal meningiomas.

### 9.2. Alternative Systemic Therapies

In addition to surgical resection and radiotherapy, other treatment modalities are being developed, which aim at targeting meningiomas through various molecular pathways [75]. Similar to the use of chemotherapy in spinal meningiomas [146,147], the vast majority of such therapies were developed to target intracranial meningiomas and are evaluated only in sporadic case reports. One phase II clinical trial examined the use of Sunitinib to target recurrent anaplastic meningiomas and reported a mean progression-free survival (PFS) of 5.2 months for all patients, with tumors expressing vascular endothelial growth factor (VEGF) faring significantly better compared to those negative for VEGF [148]. Nonetheless, that study comprised only a total of six infratentorial and spinal meningiomas, thus significantly limiting the generalizability of that trial to such lesions. Overall, there is a paucity of molecular characterization and classification of spinal meningiomas, which limits the application of known molecular agents in the treatment of these tumors. Further research is required at the molecular level to enable the molecular targeting of spinal meningiomas with alternative systemic therapies [87].

## 10. Conclusions

Spinal meningiomas are rare lesions dwarfed in frequency by their intracranial counterparts. In a modern era of burgeoning intraoperative technology and efforts to improve perioperative morbidity, maximal safe surgical resection remains the cornerstone of treatment, regardless of histologic subtype, molecular characterization, and even age. Technical challenges are associated with ossified tumors, ventral locations, and en plaque morphology, yet the majority of lesions can be effectively addressed with posterolateral decompressive approaches. Increased perioperative morbidity risk is associated with calcified spinal meningiomas and increased patient age. Adjuvant therapies, including radiotherapy and systemic medications that target the various molecular pathways implicated in spinal meningiomagenesis, remain limited and largely investigational without concrete clinical application. The extent of resection and the histological grade remain the only consistently identifiable independent predictors of survival. In order to improve our understanding of molecular tumor characterization and the utility of multimodal oncologic treatment strategies for these primary spinal tumors, future bench and bedside research efforts merit the delineation of spinal meningiomas as a separate disease entity.

## Figures and Tables

**Figure 1 cancers-16-01426-f001:**
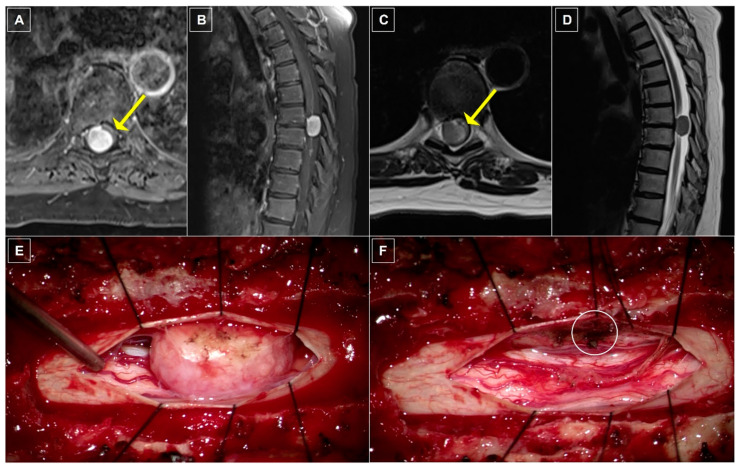
Case One: radiographic and intraoperative imaging. (**A**) Axial and (**B**) sagittal T1-weighted thoracic MRI with gadolinium contrast demonstrating an avidly, homogenously enhancing lesion nearly filling the entirety of the spinal canal with significant ventral displacement and compression of the spinal cord (yellow arrow). (**C**) Axial and (**D**) sagittal T2-weighted MRI re-demonstrating spinal cord compression (yellow arrow) as well as a tissue plane separating the spinal cord from the extramedullary mass. (**E**) Intraoperative photograph depicting a large, fleshy dorsal intradural, extramedullary mass prior to resection. (**F**) Intraoperative photograph demonstrating the extent of spinal cord compression after resection with cauterized dural tail (white circle).

**Figure 2 cancers-16-01426-f002:**
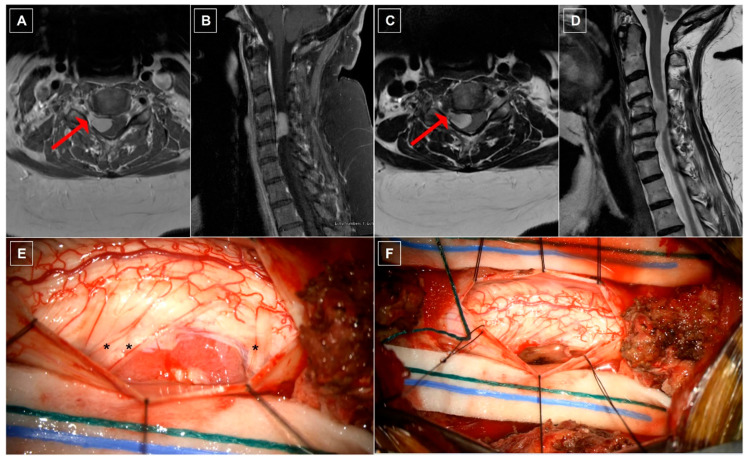
Case Two: radiographic and intraoperative imaging. (**A**) Axial and (**B**) sagittal T1-weighted cervical MRI with gadolinium contrast demonstrating an avidly, homogenously enhancing lesion ventrolateral to the spinal cord with extension into the adjacent neural foramen (red arrow). (**C**) Axial and (**D**) sagittal T2-weighted MRI re-demonstrating extension of the tumor into the right neural foramen (red arrow). (**E**) Intraoperative photograph depicting a large, fleshy ventrolateral intradural, extramedullary mass eccentric toward the right C4 and C5 neural foramina with nerve roots visibly draped over the cephalad and caudal regions of the mass (black asterisks) prior to resection. (**F**) Intraoperative photograph following surgical resection of the tumor.

**Figure 3 cancers-16-01426-f003:**
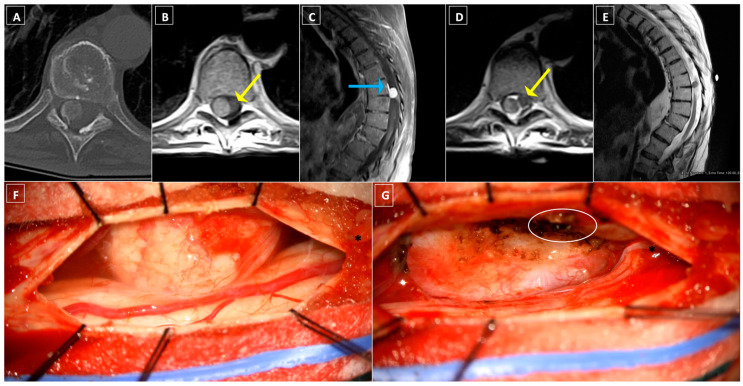
Case Three: radiographic and intraoperative imaging. (**A**) Axial CT image of the thoracic intraspinal mass with internal calcifications in the posterolateral margin of the lesion with adjacent hyperostosis of the right lamina. (**B**) Axial and (**C**) sagittal T1-weighted thoracic MRI with gadolinium contrast demonstrating a homogenously enhancing lesion lateral to the thoracic spinal cord with significant contralateral displacement of the spinal cord (yellow arrow) and dural tail (blue arrow) in a patient with Scheuermann kyphosis. (**D**) Axial and (**E**) sagittal T2-weighted MRI re-demonstrating spinal cord compression (yellow arrow) as well as a tissue plane separating the spinal cord from the extramedullary mass. (**F**) Intraoperative photograph depicting a large, fleshy dorsal intradural, extramedullary mass with nerve root tethering (black asterisk) prior to surgical resection. (**G**) Intraoperative photograph after microsurgical tumor excision demonstrating a large resection cavity and bipolar cauterization of the dural tail (white circle), with preservation of the previously tethered nerve root (black asterisk).

## Data Availability

Data supporting this review can be found within the individual articles cited in the text and provided within the bibliography.

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
