# Peer review of "Spinal Meningiomas: A Comprehensive Review and Update on Advancements in Molecular Characterization, Diagnostics, Surgical Approach and Technology, and Alternative Therapies"

_cancers, 2024, doi:10.3390/cancers16071426_

Round 1

Reviewer 1 Report

Comments and Suggestions for Authors

The authors performed a review and update on advancements in molecular characterization, diagnostics, surgical approach and technology, and alternative therapies.

Methods of the review does not appear in the manuscript.

The added value of the illustrative cases remains uncertain.  

The review widely adresses the topic of spinal meningiomas and is complete and easy to read.

Adjuvant therapies part is limited and non-sufficient.

Recently, Cancers published El-Hadjj et al. article, which limits the novelty of the present work.

Author Response

The authors performed a review and update on advancements in molecular characterization, diagnostics, surgical approach and technology, and alternative therapies.

  1. Methods of the review does not appear in the manuscript.

Author’s Reply: While this review is not a systematic review following any specific guidelines, the authors agree that a section detailing the methods for analyzing this subject matter and possible restrictions is important. A new section following the introduction has been included, entitled “2. Methods”.

Changes to Text:

  • Page 2, Lines 66-75, Methods: “A review of the scientific literature pertaining to advancements in knowledge in the following domains was performed for spinal meningiomas through July 2023: epidemiology, clinical presentation, neuroimaging, histopathology, molecular genetics, surgical treatment and clinical outcomes, and adjuvant therapies. Keywords used in multiple databases included: “spine”, “spinal”, “meningioma”, “intradural extramedullary tumor”, “clear cell meningioma”, and “primary spinal tumor”. Bibliographies of relevant literature were iteratively reviewed for inclusion until each topic was comprehensively analyzed with a focus on studies and reviews performed within the last five years. Studies were not restricted by type but were restricted to the English language.”

  1. The added value of the illustrative cases remains uncertain.  

Author’s Reply: The inclusion of illustrative clinical cases aims to (1) differentiate this comprehensive review from other previously published reviews which do not provide such examples, and (2) provide a clinical foundation for the audience to build upon and reference throughout the review, particularly as details pertain to the more common presentations and facets to this pathology. Specifically, the pre-and post-operative imaging aids readers in understanding the anatomic considerations for surgical approach and treatment and the patterns of neurovascular compression that underlie development of symptoms. Given the length of the review, we feel that the cases may be more impactful when limited to 3 cases, therefore, the manuscript has been modified to eliminate case 4.

Changes to Text:

  • Page 12, Illustrative Clinical Cases – “7.4. Case Four: Ventrolateral Thoracic Meningioma” – deleted.

  1. The review widely adresses the topic of spinal meningiomas and is complete and easy to read.

Author’s Reply: The authors thank the Reviewer for their comment regarding the comprehensive and coherent nature of this important review.

Changes to Text: None.

  1. Adjuvant therapies part is limited and non-sufficient.

Author’s Reply: One objective of this review is to highlight the paucity of validated adjuvant therapies for treatment of spinal meningiomas. Surgical resection presently remains the cornerstone of management. Development of adjuvant therapies such as chemotherapies, radiotherapy, and immunotherapy is primarily based upon the knowledge of tumor behavior pertaining to intracranial meningiomas with limited application to those presenting in the spine. As such, the authors provide a summary spanning over one page detailing some promising developments in radiotherapy and medical therapies, and represent an area to be expanded upon in future reviews once more reliable data has been obtained.

Changes to Text: None.

  1. Recently, Cancers published El-Hadjj et al. article, which limits the novelty of the present work.

Author’s Reply: El-Hajj et al. provide a fantastic, data-driven, meta-analysis of spinal meningiomas based upon established literature that was spread over two publications. Their primary points of investigation included epidemiology, tumor characteristics, non-surgical treatments, surgical treatment approach, complications, and outcomes. The objective of this submission is to provide an accessible and descriptive review of spinal meningiomas summarizing all of the same points of investigation but also with greater emphasis on neurosurgical nuances including intraoperative technical principles, challenging anatomical considerations, and illustrative clinical cases to provide a foundation of which to differentiate spinal meningiomas from their intracranial counterpart. The authors also aimed to emphasize future directions and highlight areas in need of future investigation. Since this submission is not a meta-analysis, we were able to assess and include a wider range of data, including case reports, case series, and preliminary clinical study data to provide a more comprehensive perspective on nuances of this disease entity and considerations for management, particularly regarding intraoperative surgical techniques and experimental therapies. This review’s novelty lies in its ability to succinctly summarize the peer-reviewed literature, also performed by El Hajj et al., comprehensively introduce ongoing and future targets of relevant investigation, and provide relatable clinical scenarios to demonstrate the multidisciplinary complexity of a common pathology occurring in its least common anatomic region.

Changes to Text: None.

Reviewer 2 Report

Comments and Suggestions for Authors

The authors present a review of treatment of spinal meningiomas. thereby, they focus on surgical aspects, postop follow up and further treatment as well as genetic findings. 

The manuscript is well written and is informative for the readers. I suggest to adress following aspects:

1. genetic aberrations are well known already and important for progression of meningiomas. See e.g.

Molecular biological determinations of meningioma progression and recurrence.

Linsler S, Kraemer D, Driess C, Oertel J, Kammers K, Rahnenführer J, Ketter R, Urbschat S.PLoS One. 2014 Apr 10;9(4):e94987. doi: 10.1371/journal.pone.0094987. eCollection 2014.   Correspondence of tumor localization with tumor recurrence and cytogenetic progression in meningiomas. Ketter R, Rahnenführer J, Henn W, Kim YJ, Feiden W, Steudel WI, Zang KD, Urbschat S.Neurosurgery. 2008 Jan;62(1):61-9; discussion 69-70. doi: 10.1227/01.NEU.0000311062.72626.D6.   Application of oncogenetic trees mixtures as a biostatistical model of the clonal cytogenetic evolution of meningiomas. Ketter R, Urbschat S, Henn W, Feiden W, Beerenwinkel N, Lengauer T, Steudel WI, Zang KD, Rahnenführer J.Int J Cancer. 2007 Oct 1;121(7):1473-80. doi: 10.1002/ijc.22855.   please discuss these findings and results as well.

2. please add intraop videos if possible

3. the intraoperative neuromonitoring for spinal tumors should be adressed as well

Comments on the Quality of English Language

none

Author Response

The authors present a review of treatment of spinal meningiomas. thereby, they focus on surgical aspects, postop follow up and further treatment as well as genetic findings. 

The manuscript is well written and is informative for the readers. I suggest to adress following aspects:

  1. genetic aberrations are well known already and important for progression of meningiomas. See e.g.

Molecular biological determinations of meningioma progression and recurrence.

Linsler S, Kraemer D, Driess C, Oertel J, Kammers K, Rahnenführer J, Ketter R, Urbschat S.PLoS One. 2014 Apr 10;9(4):e94987. doi: 10.1371/journal.pone.0094987. eCollection 2014. Correspondence of tumor localization with tumor recurrence and cytogenetic progression in meningiomas.Ketter R, Rahnenführer J, Henn W, Kim YJ, Feiden W, Steudel WI, Zang KD, Urbschat S.Neurosurgery. 2008 Jan;62(1):61-9; discussion 69-70. doi: 10.1227/01.NEU.0000311062.72626.D6. Application of oncogenetic trees mixtures as a biostatistical model of the clonal cytogenetic evolution of meningiomas.Ketter R, Urbschat S, Henn W, Feiden W, Beerenwinkel N, Lengauer T, Steudel WI, Zang KD, Rahnenführer J.Int J Cancer. 2007 Oct 1;121(7):1473-80. doi: 10.1002/ijc.22855. please discuss these findings and results as well.

Author’s Reply: The authors thank the Reviewer for offering several additional references regarding the molecular biological determination of meningiomatosis and how it relates to prognosis. The references articles either specifically refer to intracranial meningiomas or do not distinguish between spinal and intracranial tumors in their study. One emphasis of this work is that much of the data driving therapeutic management of spinal meningiomas is based on treatment paradigms assessed in intracranial meningiomas and mechanistic work utilizing cell lines derived by intracranial meningiomas. There is a paucity of literature that assess genetic similarities between spinal meningiomas and their counterpart. Accordingly, it is unclear how generalizable this data is with respect to spinal meningiomas. As such, the manuscript will be modified to include these citations in the respective areas that discuss known molecular determinants for meningiomas in general, however, further discussion of these specific studies should be avoided to maintain emphasis on genetic aberrancies known to spinal meningiomas.

Changes to Text:

  • Page 5, Line 263, Molecular Genetics: “chromosome 1p and/or 9p alterations.”
  • Page 6, Lines 309-313, Molecular Genetics: “Finally, while molecular characterization has proven pertinent to the biological determination of meningioma progression and recurrence, some studies still suggest that factors related to surgery rather than genomic profiling confer the greatest contribution to recurrence, a supposition that again derives from intracranial meningioma research.”
  • References: 2 citations added
    • Linsler et al 2014. Doi: 10.1371/journal.pone.0094987.
    • Ketter et al 2007. Doi: 10.1002/ijc.22855

  1. please add intraop videos if possible

 Author’s Reply: Unfortunately, no videos of our spinal meningioma surgeries are available.

Changes to Text: None.

  1. the intraoperative neuromonitoring for spinal tumors should be adressed as well

Author’s Reply: We thank the Reviewer for agreeing that intraoperative neuromonitoring is important to consider when discussing the surgical advancements for spinal tumor resection. This topic was discussed in section 6, “Surgical Resection” (Lines 295-301) whereby we state that there may be favorable sensitivity and specificity for identification of intraoperative iatrogenic injury, however, evidence for the use of neuromonitoring for spinal meningiomas specifically is limited. No studies at present demonstrate improvement in postoperative outcomes following its use. We hope that the illustration of this critical point will inspire further research on the topic.

Changes to Text: None.

Round 2

Reviewer 2 Report

Comments and Suggestions for Authors

The Comments are adressed. Accept as it is 
